# Deliberate Exposure to Opposing Views and its Association with Behavior and Rewards on Political Communities

## ABSTRACT

Engaging with diverse political views is important for reaching better collective decisions, however, users online tend to remain confined within ideologically homogeneous spaces. In this work, we study users who are members of these spaces but who also show a willingness to engage with diverse views, as they have the potential to introduce more informational diversity into their communities. Across four Reddit communities (*r/Conservative, r/The_Donald, r/ChapoTrapHouse, r/SandersForPresident*), we find that these users tend to use less hostile and more advanced and personable language, but receive fewer social rewards from their peers compared to others. We also find that social sanctions on the discussion community *r/changemyview* are insufficient to drive them out in the short term, though they may play a role over the longer term.

## 1 INTRODUCTION

Social media can widen democratic participation and promote information exchange [27]. However, they may also absorb users into online groups, potentially giving rise to uncivil interactions dominated by a select few users [17].

Constructive or deliberative interactions between people with diverse views can lead to higher-quality information exchange, even when such interactions are competitive [37, 38, 47]. However, online interactions mostly occur with homogeneous ideas and users (echo chambers) [11, 41, 44, 50], and when they do happen between users of opposing ideologies, they tend to be negative and unconstructive [5, 10, 28]. Moreover, political disagreements may lead users to disengage from politics [42] or to seek out views that reaffirm their initial beliefs [7, 45].

Users who are part of such homogeneous online groups but who otherwise demonstrate a willingness to partake in heterogeneous discussions can be a promising avenue for bringing new ideas into these groups, yet this remains understudied. Here, we set out to better understand these users and how they are treated by their communities. We operationalize such users as those who post or comment in the *r/changemyview* (CMV) subreddit, i.e., users who *deliberately* seek out and engage with opposing views, but who are also active in political subreddits with more defined ideological alignments. Specifically, we pose the following research questions:

**RQ1** Do CMV participants receive fewer social rewards (i.e., net upvotes) in their home communities than non-participants?

**RQ2** What are the differences in the language used between CMV participants and non-participants, if any?

**RQ3** Can social punishments (i.e., downvotes) in discussion communities drive users out of these communities and into seclusion?

*WWW '24, May 13–17, 2024, Singapore*
2024. ACM ISBN 978-1-4503-XXXX-X/18/06...$15.00
https://doi.org/XXXXXXX.XXXXXXX

**r/changemyview.** CMV has been described as an "anti-echo chamber" [15]. Users make submissions asking other users to present arguments against an opinion they hold in the comments. Arguments must be made genuinely, and posting users must truly be willing to change their view. The community is heavily moderated for civility and engaging in good faith, and is in the top 150 most subscribed subreddits.[1] We rely on this community for our methodology as it characterizes a group of users who *deliberately* expose themselves to diverse views, either by inviting them or counter-arguing them, in a *civil* and *genuine* manner.

**Methodology.** To address **RQ1**, we obtain data from four political Reddit communities (a.k.a., *subreddits*), which lie between the far left and far right of the political spectrum (*r/ChapoTrapHouse; CTH, r/SandersForPresident; SFP, r/Conservative; CON, r/The_Donald; TD*). In Section 4, we allocate each user to one of the four subreddits based on commenting activity. Then, from each community, we subset users who also participate in CMV, i.e., those who actively seek out opposing views. We match them to other similar users in their community and compare the net upvotes that they receive.

For **RQ2**, we analyze linguistic differences between CMV participants and other users in Section 5. We examine their language's grade level using readability formulas, hostility as determined through Perspective API models, psychological traits using LIWC-22, and entities and topics they discuss.

For **RQ3**, we utilize a full year of data for *all* users who appear in CMV (10.1M comments from 76.8K users) in Section 6. We obtain their user trails, looking at which subreddits they comment on and their comments' scores. Using higher-order Markov chains, we compute transition probabilities from certain communities to others, given their history's subreddits and scores.

**Main findings.** Overall, we find that CMV participants receive 4.34% to 10.15% fewer upvotes than non-participants in their home communities. They also differ in linguistic style, using higher-grade text, less hostile and confident but more personable and authentic language, and discussing slightly different topics.

Social sanctions are not enough to drive users out of CMV in the short term. However, those who stay in the discussion community over the longer term attract fewer downvotes than users who eventually leave.

Our findings have several implications. First, though accepted by their communities, users with diversified exposure are not as popular. Thus, harnessing their openness by making their voices more prominent within their own spaces is an open challenge. Second, their language is more moderate, which could be linked to their lower popularity; efforts to make such language more normative over time may be fruitful. Finally, although disapproval does not drive users away in the short term, it may need to be balanced over the longer term to encourage continued engagement.

[1] https://www.reddit.com/best/communities/1/

## 2 RELATED WORK

In this section, we cover work on why users are socially rewarded and the role of such rewards on engagement. Moreover, we look at the type of content that is preferred by denizens of online spaces.

**Who is rewarded online?** On Twitter, partisan content receives more engagement than bipartisan or neutral content [12]. On the far-right, pro-Trump Reddit community *r/The_Donald*, the 1000 most upvoted comments in 2017 featured substantially more extreme and hateful speech than less-upvoted comments [13]. Furthermore, interviews with moderators of the Reddit community *r/AskHistorians*, which aims to provide accurate descriptions of historical events, reveal that visitors of the community tend to mostly upvote comments which seem attractive and align with their biases, while more accurate comments receive fewer upvotes [14]. On the other end, users who eventually leave the conspiracy-minded QAnon community begin to receive lower net scores on their dissenting comments leading up to their departure [34].

**The role of rewards on engagement.** Users may look at their own comments' scores to gauge support from their community [21], therefore, these scores can affect their engagement. Some Reddit users express negative sentiment after being downvoted, but positive sentiment after being upvoted [9]. Surveys reveal that comment score and user status are motivating factors behind why Reddit users may choose to participate in discussions on the platform [30].

Scores may also affect engagement on social Q&A sites, e.g., Stack Exchange. Upvotes on answers are linked with more subsequent contributions [46], while new users may decide whether to continue participating in such sites based on the scores that their questions receive [23]. However, in some situations, the opposite effect holds; upvotes may reduce contributions, perhaps because users do not want to risk their good reputation, while downvotes may motivate users to improve their scores by engaging more [31].

**Current literature gaps.** Existing work demonstrates that more neutral or disagreeable users receive fewer rewards from others [12, 34], and how such rewards or sanctions motivate users' engagement [9, 30]. Yet, it remains unclear whether members of communities with specific narratives but who are otherwise willing to engage with broader views through good-faith discussions are penalized by their peers.

These users are important to understand, as they may be uniquely positioned to bring more diverse ideas into their communities or normalize more open-minded language [8]. This is especially pertinent given that people are often secluded in specific groups [50] or turn hostile when engaging with other-minded people [10, 28], thus making them apprehensive of influence from "outsiders" [49]. At the same time, it is worth studying whether social rewards can affect engagement even in communities that are specifically designed for wide-ranging discussions (and thus run a high risk of encountering disagreement), as this may carry implications for how online deliberations are enacted. In this paper, we aim to address both of these gaps.

## 3 DATASET

We obtain data from a far-left (*r/ChapoTrapHouse*; CTH), a moderate-left (*r/SandersForPresident*; SFP), a moderate-right (*r/Conservative*; CON), and a far-right (*r/The_Donald*; TD) subreddit, using the

| Dataset | Sub | #comments | #authors | Dates |
|---|---|---|---|---|
| A | TD | 37.7M | 545K | |
| | CTH | 7.00M | 108K | |
| | CON | 2.17M | 140K | Jul 16-Dec 19 |
| | SFP | 1.31M | 146K | |
| | **total** | **48.2M** | **799K** | |
| B | CMV | 10.2M | 76.9K | Jan-Dec 18 |

**Table 1: Data description.**

Pushshift API [3]. CTH and TD were banned in 2020 for Reddit rule violations, including promoting violence. We collect data between July 20th, 2016, which is the creation of the youngest subreddit among the four (CTH), and December 31st, 2019. We choose these four subreddits because preliminary analyses revealed that they are all among the top 20 political subreddits[2] in terms of participating users, and they all have specific accepted narratives. TD and SFP advocate for Donald Trump and Bernie Sanders, while CON and CTH espouse conservatism and anti-capitalism, respectively.

In addition, we collect one year of data for all users who appear in CMV between January 1st and December 31st. Table 1 is a description of these datasets.

## 4 PENALTIES TO CMV PARTICIPANTS

In this section, we compare comment scores between users who deliberately expose themselves to opposing views (i.e., CMV participants) and others. We treat any user who has made at least one comment *or* submission on CMV as a participant, and a non-participant otherwise. Although submissions and comments differ in nature, both involve good-faith discussions between different-minded people and both pertain to active engagement with opposing views.

### 4.1 User Allocation and Matching

Following previous work [1, 11, 36], we assign users to one of the four subreddits as their "home" if, within our data pool, they have the majority of their comments *and* an overall score above 1 (Reddit's default comment score) there.

To obtain comparable *case* (i.e., CMV participant) and *control* (i.e., non-participant) groups, we follow a similar matching approach to Phadke et al. [35]. First, we subset all users per subreddit who *also* appear in CMV. This forms our four case sets. All other home users for each subreddit are potential controls for the respective subreddit's case set. We then match cases to potential controls on 7 features: total comments during the 1) observation and 2) pre-observation period, proportion of comments in their home subreddit during 3) observation and 4) pre-observation, total subreddits commented in during 5) observation and 6) pre-observation, and 7) date of their first comment (to the nearest day). We only consider activity on political subreddits [36].

Based on these features, we conduct nearest-neighbor Mahalanobis distance matching with replacement. We remove pairs which contain bots (see Appendix A), although sensitivity analyses reveal almost identical results when including bots. To assess matching robustness, we obtain Standardized Mean Differences (SMDs) between the case and control sets for each subreddit and

---

[2]We define a political subreddit as any subreddit in which 50% or more of its comments are political, as determined by prior work [36].

across each matching feature. SMDs below 0.20-0.25 [24, 35, 40] generally indicate good matching. No SMD exceeds 0.15 for any feature, indicating robust matching (Table 2).

## 4.2 Validation Study

Before proceeding, we validate the meaningfulness of the distinction between CMV participants and non-participants in each subreddit. We follow a method by Garimella et al. [12], who assess users' degree of partisanship by analyzing the political leaning of news sources that they share as labelled by Bakshy et al. [2]. Since CMV participants should be less partisan (more open-minded), we expect that they will share news in a less one-sided manner.

Bakshy et al. [2] assign model predictions between -1 (fully left) and 1 (fully right) to the top 500 domains on Facebook in the first half of 2014. We reinforce this with a more recent 2019 dataset, which presents bias ratings for 548 sources labeled by human assessors from AllSides Media Bias. Ratings have 5 categories, from -2 (very left) to 2 (very right), with 0 indicating center. We transform the Bakshy et al. dataset into this categorical scale by splitting the continuous scores into 6 even bins (and taking the two middle bins to be center). For domains that are in both datasets but have different scores, we keep the human-assessed AllSides score. Excluding duplicates, we consider 795 domains.

We extract URLs from every comment that users post in any subreddit across the observation period, and take the mean domain score across all comments for each group per subreddit. We also do this for fully random samples of users as an additional validation layer ($N_{random} = N_{cases}$). As expected, cases are less biased than controls and random users in every subreddit; that is, left-leaning cases (CTH and SFP) post less left-leaning, and right-leaning cases (CON and TD) post less right-leaning URLs than control or random users (Table 3). This also holds when measuring bias at the user, instead of the comment level.

Note that the vast majority of the top 12 most-visited outlets [32] receive a left-leaning score (CNN, NYT, NBC/MSNBC, WaPo, BuzzFeed, CBS, ABC, HuffPost), with only one (Fox) being right-leaning and three (NPR, USAToday, MSN) being center. This might explain the seemingly paradoxical, near-neutral score of TD cases and CON controls, and the left-leaning score of CON cases (i.e., they likely simply share more mainstream sources than others). Due to our user matching, controls are also more moderate than random samples. Overall, this check validates that CMV participants are less partisan than controls or random users.

## 4.3 Differences in Rewards

Next, we compare the average scores between cases and controls for each subreddit. For robustness, we perform both parametric (value-based) and non-parametric (rank-based) comparisons. For non-parametric inference, we perform Mann-Whitney U tests. In cases of violation of the equality of variances assumption (i.e., significant Levene's tests), we corroborate this with a further median test.

We show non-parametric results in Table 4. Control users have significantly higher mean ranks than case users in all subreddits meaning that they receive higher scores, except for CON, which is marginally non-significant. However, despite statistically significant differences in 3 out of 4 subreddits, all effect sizes are very small (Cohen's $d < 0.2$).

For parametric inference, we use Welch's t-tests because equality of variances is violated in all cases. All groups show normality violations (significant one-sample Kolmogorov-Smirnov tests against a normal distribution), although the group sizes should be robust against these. Normality violations persist through log-transformations; thus, we instead trim the top 5% values as outliers. We show parametric results in Table 5, which corroborate our non-parametric tests. Controls have significantly higher mean upvotes per comment compared to cases in all subreddits. Once again, the effect sizes are small. These differences remain significant in sensitivity checks without outlier removal, except CON.

Altogether, CMV participants receive fewer social rewards than non-participants. The effect sizes are small, but they are comparatively higher in the left-leaning than in the right-leaning subreddits.

## 5 LINGUISTIC DIFFERENCES

Next, we analyze linguistic differences between CMV participants and non-participants. Specifically, we look at: 1) ease of readability, 2) hostility (toxic, profane, or insulting language), 3) psychological traits (analytical, authentic, and confident language), 4) named entities, and 5) topics.

## 5.1 Readability

Readability tests return a US school grade level (e.g., 4, 5), denoting that the text would be easily understood by an average 4th, 5th, etc. grader. Most require at least 100 words per text, thus, we filter out comments with fewer words than this; we show how many are retained in Table 6. In all subreddits, case users post more long texts than control users, both in absolute and relative terms.

Using Python's textstat library, we obtain a readability grade level for each comment using 8 different formulas (see Appendix B). We then take the modal grade level across all formulas for each comment. As this data is ordinal, we use Mann-Whitney U tests to assess differences between the ranks of cases and controls, with further median tests where equality of variances is violated.

The mean rank of cases is significantly higher than the mean rank of controls for all four subreddits, suggesting that comments made by CMV participants are overall more difficult to read in terms of grade level (Table 7). Again, all effect sizes are fairly small, but they are comparatively larger for the non-extreme subreddits (CON and SFP).

## 5.2 Hostility Attributes

Our next analysis focuses on whether non-participants use more hostile language than CMV participants. Specifically, we examine whether there are significant differences in the proportion of toxic, insulting, or profane comments between case and control groups for each subreddit. To that end, we use three models from Google Jigsaw's Perspective API [22]:

(1) *Severe Toxicity*, defined as "a very hateful, aggressive, disrespectful comment or otherwise very likely to make a user leave a discussion or withhold their perspective."
(2) *Insult*, defined as an "insulting, inflammatory, or negative comment towards a person or a group of people."
(3) *Profanity*, defined as "swear words, curse words, or other obscene or profane language."

| Subreddit | 1st comment | Standardized Mean Difference | | | | | | | |
|---|---|---|---|---|---|---|---|---|---|
| | | Observation | | | Pre-Observation | | | | |
| | | #comments | %home | #subs | #comments | %home | #subs | #cases | #controls |
| r/The_Donald | -0.03 | 0.05 | -0.03 | 0.11 | 0.03 | 0.02 | 0.05 | 19948 | 17640 |
| r/Conservative | -0.03 | 0.06 | 0.01 | 0.15 | 0.03 | 0.02 | 0.07 | 8854 | 7231 |
| r/SandersForPresident | -0.04 | 0.08 | 0.04 | 0.14 | 0.05 | 0.02 | 0.08 | 8026 | 6679 |
| r/ChapoTrapHouse | -0.04 | 0.05 | -0.02 | 0.11 | 0.03 | N/A | 0.07 | 6251 | 5339 |

Table 2: SMDs for every feature across all subreddits. Notice the N/A value in home comments for the pre-observation period in r/ChapoTrapHouse because the subreddit did not exist then.

| Subreddit | $M_{case}$ | $M_{control}$ | $M_{random}$ |
|---|---|---|---|
| TD | -0.03 | 0.32 | 0.56 |
| CON | -0.27 | -0.01 | 0.18 |
| SFP | -0.73 | -0.78 | -0.80 |
| CTH | -0.79 | -0.84 | -0.86 |

Table 3: Mean source bias per group per subreddit.

Although Perspective is sensitive to adversarial text [20], it still outperforms alternative models [48]. The models return values ranging from 0 to 1. We use a cutoff of $\geq 0.8$ for all models, which is adequately high to avoid false positives [19, 25].

With this classification, we obtain one contingency table per attribute for each subreddit (12 tables in total), on each of which we perform chi-square tests (Table 8). We apply Bonferroni corrections as we make 3 comparisons with each population; thus, we interpret significance at $p = 0.017$.

In 9 out of 12 comparisons, we detect significantly more hostile language in controls compared to cases. The opposite holds for profanity in CTH; CMV participants in this subreddit use more swear words. CON's and TD's controls are more hostile than cases in all attributes and also show the biggest differences overall. Every attribute is more frequent among control users in 3 out of 4 subreddits.

## 5.3 Psychological Traits in Language

Next, we examine psychological dimensions exhibited in users' language using three of LIWC-22's [6] summary metrics (Analytic, Authentic, Clout). Analytic reflects formal and logical language; Authentic reflects the degree to which the user avoids adjusting language to fit their social environment; and Clout reflects the confidence and social status expressed in the user's writing [33]. As LIWC relies on a dictionary approach, we filter out comments with fewer than 10 words in these analyses. For robustness, we perform both parametric and non-parametric comparisons of text scores between cases and controls for all three traits, which we show in Table 9.

Once again, we find consistent patterns across all four subreddits. As expected, control users are higher in Clout (all $p < 0.001$), which indicates that they express more confidence and social status in their comments. However, contrary to what one might expect, control users are also higher in their use of Analytic language (all $p < 0.001$). At the same time, case users use more Authentic language (all significant at the Bonferroni-corrected $p < 0.017$ cutoff, with the exception of SFP's parametric corroboration).

Taken together, the LIWC analysis reveals that CMV participants may not adjust their language to suit their social environment as much as non-participants do; this could partly explain the comparably fewer social rewards they garner, as self-monitoring in this way is associated with better impression management and likeability [43]. At the same time, non-participants write with more confidence, which can also lead to more positive evaluations by others [29]. Though we expected CMV participants to also use more analytical language, we observed the opposite effect. Thus, they may use more personable language compared to the more formal language of non-participants, perhaps because formal language is more normative within the political spaces we study.

In order to get more context around these discussions, our next analyses focus on entities and topics mentioned.

## 5.4 Named Entity Recognition (NER)

For NER, we use the pre-trained en_core_web_trf transformer model based on the RoBERTa architecture [26] from Python's spacy library. To improve suitability, we annotate 5K random comments from political subreddits [36] and train a new model on top of the pre-trained one with an 80-10-10 train-validation-test split.

This new model captures more informal terms (e.g., "dems", "neocons", etc.), provides increased performance against the test set (F1 score of 77.18 vs 48.62 of the pre-trained model), and includes two more entity types that we added (Ideology and Website).

We show all entity types and brief definitions in Table 10, along with any amendments we make relative to the pre-trained model. We perform NER on all 7.51M comments made by case and control users,[3] and plot the *relative* prominence of each entity in Figure 1.

Organizations and persons are mentioned more by control users than case users in all subreddits. Ordinal and cardinal numbers, percentages, laws, and geopolitical entities are universally mentioned more by case users. Generally, controls refer *slightly* more to personified entities, while cases refer more to numeric and legal entities. Nonetheless, these patterns are very subtle; overall, they discuss similar kinds of entities. This also holds for the exact entities mentioned per type, shown in an anonymous Google Sheet[4].

## 5.5 Topic Extraction

Next, we compare the topics that cases and controls discuss in each subreddit using Latent Dirichlet Allocation (LDA) [4]. We remove URLs and stopwords, lemmatize the text, extract bigrams, and apply Term Frequency-Inverse Document Frequency weights.

---

[3]We observe similar patterns using the pre-trained model.
[4]https://docs.google.com/spreadsheets/d/e/2PACX-1vSzW-eubTn2GZpAYDFuzjHJh3mcfICoJdH7qjrBvTKqPWNsHrhr V44rudVGJ3RB4A/pubhtml

| | | Case | | Control | | Mann-Whitney | | Median test | |
|---|---|---|---|---|---|---|---|---|---|
| Subreddit | EoV | Median | Mean rank | Median | Mean rank | Cohen's d | p | Stat | p |
| r/The_Donald | False | 3.91 | 18375 | 4.11 | 19269 | 0.082 | < 0.0001 | 41.43 | < 0.0001 |
| r/Conservative | True | 5.14 | 7979 | 5.40 | 8121 | 0.030 | 0.054 | N/A | N/A |
| r/SandersForPresident | False | 4.00 | 7171 | 4.53 | 7571 | 0.094 | < 0.0001 | 23.40 | < 0.0001 |
| r/ChapoTrapHouse | True | 9 | 5567 | 10.27 | 6064 | 0.148 | < 0.0001 | N/A | N/A |

**Table 4: Medians, mean ranks, and statistics for non-parametric tests of upvotes by group. EoV is True if the equality of variances assumption is met and False otherwise, in which case we follow the Mann-Whitney test up with a median test.**

| Sub | $M_{case}$ | $M_{control}$ | %diff | t | Cohen's d |
|---|---|---|---|---|---|
| TD | 4.81 | 5.04 | 4.67 | **-5.87 | 0.061 |
| CON | 6.54 | 6.83 | 4.34 | *-3.15 | 0.05 |
| SFP | 5.75 | 6.20 | 7.53 | **-4.66 | 0.077 |
| CTH | 9.91 | 10.97 | 10.15 | **-7.54 | 0.141 |

**Table 5: Means and parametric test statistics of upvotes by group. $^*p < 0.01$, $^{**}p < 0.0001$.**

| | Case | | | Control | | |
|---|---|---|---|---|---|---|
| Sub | #ret | #total | %ret | #ret | #total | %ret |
| TD | 88.2K | 2.31M | 3.82 | 73.2K | 2.51M | 2.92 |
| CON | 30.3K | 329K | 9.21 | 19.6K | 241K | 8.12 |
| SFP | 18.5K | 152K | 12.14 | 9.85K | 104K | 9.48 |
| CTH | 47.2K | 948K | 4.98 | 40.0K | 919K | 4.34 |
| ovrl | 184K | 3.74M | 4.91 | 143K | 3.77M | 3.79 |

**Table 6: Absolute (#) and relative (%) numbers of retained (ret) texts (≥ 100 words) for readability analysis.**

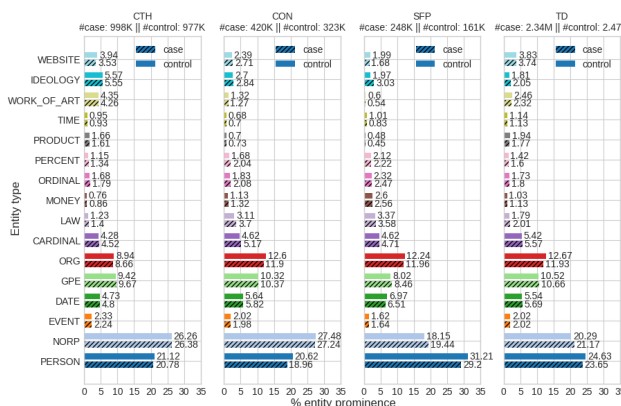

**Figure 1: Prominence comparison of all entity types by subreddit group. #case and #control indicate total number of entities detected across all comments. Fac, Language, Quantity and Loc omitted due to < 1% prominence in all groups.**

Then, we iterate the number of topics hyperparameter from 5 to 15 and extract the number which produces the highest coherence score for each group (ranging from 7 to 15 topics). Due to space constraints, we only show our *interpretations* for the top 10 topics

extracted per group in Table 11. We present the full topics (and constituent words) in an anonymous Google Sheet.[5]

The topic analysis affords a more grounded understanding of users' discussions. In the case of CTH, cases do not stray too far from political subjects. On the other hand, controls additionally discuss art forms like movies and podcasts (including other controversial left-associated podcasts), while also hinting towards emotional states like disdain towards critics. SFP users are more expressly political, although control users also veer into more abstract concepts such as voicing doubts and corruption concerns mostly about the right. In TD, topics often concern "others"; however, "others" mostly mean political opponents with case users (e.g., topic 5 which refers to Hilary Clinton), while they refer to other religions and countries with control users (e.g., topics 5 and 8). With CON, we see that case users pick up on more narrow conservative talking points (abortion, guns, justice system), whereas controls adopt a more general view (e.g., right vs. wrong in topic 3 and conservative values in topic 7) with the exception of topic 6 on immigration specifically. TD and CTH topics feature substantially more profanity than the more moderate subreddits. Overall, differences between case and control users range from an explicit topical focus on politics to the topics' level of abstraction.

## 5.6 Inferences from Linguistic Analyses

Overall, these analyses show that CMV participants use slightly more advanced and less hostile language, while also being friendlier and more authentic; non-participants are more formal and confident. While differences in entities discussed are miniscule, CMV participants' topics seem to be less abstract and more explicitly political.

## 6 PERSISTENCE AGAINST DOWNVOTES

Next, we address whether downvotes can drive users out of discussion communities to seek approval elsewhere, e.g., in their home communities. We obtain all comments between January 1st and December 31st, 2018, for any user who posted at least one comment in CMV during this period (excluding bots). This yields 10.1M comments made by 76.8K authors across 792 subreddits (including CMV).

We build trails for every user and split these into separate sessions when over 8 hours elapse between two successive comments. For robustness, we also split at 4, 12, and ∞ hours (i.e., no splits). We note each comment's subreddit (CMV, home, or other) and score

---
[5]https://docs.google.com/spreadsheets/d/e/2PACX-1vRvanN-nTJ-DesPWxLmk56OjAOoQFNUPTKccFx35dobIGaEVjzstUzUt9ae2XaNEA/pubhtml

| Subreddit | EoV | Case | | Control | | Mann-Whitney | | Median test | |
|---|---|---|---|---|---|---|---|---|---|
| | | Median | Mean rank | Median | Mean rank | Cohen's $d$ | $p$ | Stat | $p$ |
| r/The_Donald | True | 9 | 80966 | 9 | 80412 | 0.012 | 0.016 | N/A | N/A |
| r/Conservative | False | 9 | 25405 | 9 | 24211 | 0.081 | < 0.0001 | 59.31 | < 0.0001 |
| r/SandersForPresident | False | 9 | 14441 | 9 | 13629 | 0.095 | < 0.0001 | 39.17 | < 0.0001 |
| r/ChapoTrapHouse | False | 11 | 44276 | 10 | 42682 | 0.063 | < 0.0001 | 57.33 | < 0.0001 |

Table 7: Medians, mean ranks, and statistics for non-parametric tests of readability by group. EoV is True if the equality of variances assumption is met and False otherwise, in which case we follow the Mann-Whitney test up with a median test.

| Sub | Attr | $\chi^2$ | ↑group |
|---|---|---|---|
| CTH | Insult | ***22.39 | Control |
| | Profanity | **14.64 | Case |
| | Toxicity | 1.43 | N/A |
| CON | Insult | ***1558.71 | Control |
| | Profanity | ***685.90 | Control |
| | Toxicity | ***153.55 | Control |
| SFP | Insult | 0.06 | N/A |
| | Profanity | **9.39 | Control |
| | Toxicity | ***24.04 | Control |
| TD | Insult | ***490.49 | Control |
| | Profanity | ***684.63 | Control |
| | Toxicity | ***725.52 | Control |

Table 8: Chi-square results for Perspective attributes per subreddit. ***$p < 0.0001$, **$p < 0.017$, *$p < 0.05$. ↑ indicates which group has a higher observed frequency of the attribute compared to the expected frequency. N/A = non-significant.

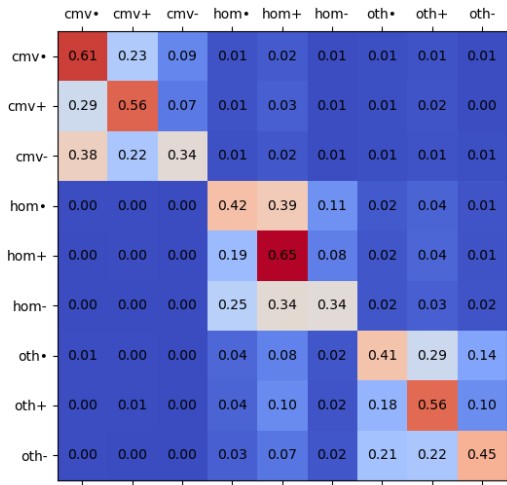

Figure 2: First-order transition matrix with 8-hour interruptions. +, -, and ● show upvoted, downvoted, and neutral comments, respectively.

(downvoted, upvoted, or neutral). "Home" here means the top 3 subreddits where the user has most of their comments, as we find that ~99% of users have at least 50% of their comments in ≤ 3 subreddits.

This creates 9 possible subreddit-vote combinations, which we treat as *states* in a higher-order Markov chain. Between 1 - 8, we find the lowest (optimal) Bayesian Information Criterion (BIC) at the 4th order. However, we also conduct analyses at the 3rd order as a sensitivity check. We only retain user trails of size $\geq N + 1$, where $N = order$.

Regardless of score, users are always most likely to stay within their current community. For example, in the 3rd-order chain, the 27 histories with the highest probability of resulting in any given community are the 27 possible combinations of comment votes *in that community*, which yield recurrence probabilities between 91.1% and 97.8%. In fact, 3 consecutive downvotes result in 0.4%, 0.5%, and 2.6% *higher* probabilities of staying in CMV, home, and other, respectively, than 3 consecutive upvotes. Figure 2, which is the first-order transition matrix, demonstrates this tendency.

This pattern is also reflected in Figure 3, where we simulate the average user's trail over 1000 comments starting with a random history and plot the "decomposed" resulting states. While we observe high fluctuations in votes, there are no commensurate (lagged) fluctuations for communities. The relatively long flat periods indicate that commenters tend to stay in the same community, regardless of votes.

Overall, this analysis suggests that disagreement or social punishments are not enough to drive users away from conversations. Instead, users mostly "stand their ground." This seems to apply even

when a user's very first comment in the community is downvoted (see Appendix C.1). However, downvotes may still play a partial role over the longer term (see Appendix C.2).

## 7 DISCUSSION AND CONCLUSION

In this work, we provide a deeper understanding of users who expose themselves to diverse views. These users are a potential avenue for introducing new ideas into communities with established narratives, therefore, we argue that they are important to study. Specifically, we examine their treatment (sanctions and rewards) by their own communities, how this may be related to the language they use, as well as whether sanctions and rewards play a role in their engagement with discussion communities themselves.

### 7.1 Implications

Here, we reiterate our findings and explain their implications.

**Rewarding bias.** To answer **RQ1**, CMV participants receive fewer social rewards than non-participants, which contextualizes previous findings around users' preference for extremity [13] and partisanship [12]. This suggests that communities prefer users who fully comply with established narratives, as their comments may more adequately satisfy the wider communities' biases [14].

It is important to reiterate that CMV participants are not *punished* by their communities, but rather, simply receive fewer rewards.

| sub | #cases | #controls | trait | $M_{case}$ | $M_{control}$ | $Mdn_{case}$ | $Mdn_{control}$ | $t$ (d) | $\Delta M_{rank}$ (d) |
|---|---|---|---|---|---|---|---|---|---|
| TD | 1.45M | 1.51M | Analytic | 42.65 | 43.97 | 38.91 | 39.7 | ***-34.21 (0.04) | ***-32.9K (0.038) |
| | | | Authentic | 38.43 | 37.85 | 26.78 | 24.32 | ***13.87 (0.016) | ***18.7K (0.022) |
| | | | Clout | 47.87 | 48.96 | 40.06 | 40.06 | ***-25.94 (0.03) | ***-25.9K (0.03) |
| CTH | 609K | 578K | Analytic | 43.97 | 45.39 | 39.7 | 42.89 | ***-23.21 (0.043) | ***-14.4K (0.042) |
| | | | Authentic | 39.9 | 39.74 | 30.98 | 30.98 | **2.43 (0.004) | **2.11K (0.006) |
| | | | Clout | 43.93 | 44.18 | 40.06 | 40.06 | **-3.72 (0.007) | ***-2.47K (0.007) |
| CON | 251K | 185K | Analytic | 42.3 | 43.19 | 38.85 | 39.7 | ***-9.1 (0.028) | ***-3.5K (0.028) |
| | | | Authentic | 39.65 | 38.46 | 30.98 | 28.56 | ***11.13 (0.034) | ***4.84K (0.038) |
| | | | Clout | 45.33 | 47.05 | 40.06 | 40.06 | ***-16.18 (0.05) | ***-6.17K (0.048) |
| SFP | 123K | 79.6K | Analytic | 41.57 | 43.34 | 37.99 | 39.7 | ***-12.42 (0.056) | ***-3.23K (0.054) |
| | | | Authentic | 38.41 | 38.85 | 29.13 | 28.56 | 0.68 (0.003) | **700 (0.012) |
| | | | Clout | 45.52 | 46.39 | 40.06 | 40.06 | ***-5.58 (0.025) | ***-1.38K (0.023) |

**Table 9: Descriptive and inferential statistics for LIWC analyses. Where equality of variances is violated, $t$ is obtained using a Welch test. #cases and #controls refer to the number of comments retained in analyses per group. $\Delta M_{rank}$ = difference in mean ranks used in non-parametric Mann-Whitney U tests. \*\*\****p* < 0.0001, \*\****p* < 0.017, \**p* < 0.05.**

| Label | Definition | Label | Definition |
|---|---|---|---|
| **CARDINAL** | Numbers that do not fall under another type | **DATE** | Absolute or relative dates or periods |
| **EVENT** | Named hurricanes, battles, wars, sports events, etc. | **FAC** | Facilities like buildings, airports, highways, bridges, etc. |
| **GPE** | Geopolitical entities (countries, cities, states) | **LANGUAGE** | Any named language |
| **ORG** | Organizations like companies, agencies, institutions, etc. | **LOC** | Non-GPE locations, mountain ranges, bodies of water |
| **LAW†** | Named documents made into laws, *or any other official government documents like reports, proposed policy, etc.* | **NORP†** | Nationalities or religious or political groups, *or ethnic, racial, or ideological groups* |
| **ORDINAL** | "First", "second", etc. | **MONEY** | Monetary values, including unit |
| **PERCENT** | Percentage, including % | **PERSON** | People, including fictional |
| **PRODUCT** | Objects, vehicles, food, etc. (not services) | **QUANTITY** | Measurements, as of weight or distance |
| **TIME** | Times smaller than a day | **WORK_OF_ART** | Titles of books, songs, movies, shows, etc. |
| **IDEOLOGY\*** | Political, economic, religious, or philosophical ideology, school of thought, or system | **WEBSITE\*** | Any named website which is not referred to in the context of an organization (e.g., Reddit) |

**Table 10: All types of entities in the model. †The entity's definition has been amended relative to the base model, with the amendment shown in italics. \*The entity was added to the trained model and does not appear in the pre-trained one.**

| sub | group | topic 0 | topic 1 | topic 2 | topic 3 | topic 4 | topic 5 | topic 6 | topic 7 | topic 8 | topic 9 |
|---|---|---|---|---|---|---|---|---|---|---|---|
| CTH | **case** | *und* | *und* | voting | far left | upset | liberals | politics | israel | society | zizek |
| | **control** | *und* | *und* | economy | obama | *und* | disdain | america | podcasts | movies | mockery |
| SFP | **case** | sanders | posting | medcare | election | ideology | right | russia | – | – | – |
| | **control** | sanders | primary | election | medcare | socialism | voting | companies | right | doubt | corruption |
| TD | **case** | *und* | posting | race | america | voting | emails | corruption | govt | trump | election |
| | **control** | *und* | *und* | memes | election | mockery | islam | trumpism | opposition | non-us | mobilize |
| CON | **case** | *und* | govt | abortion | voting | politics | internet | reddit | guns | justice | media |
| | **control** | voting | govt | taxes | morality | satire | media | us border | con values | – | – |

**Table 11: Topic interpretations by group. *Und* stands for undefined; most of these topics reflect colloquial exchanges, i.e., vague words (e.g., thanks, please, good) and/or profanity. "Politics" = references to *both* the left and right. "Posting" = online activity. "Right" refers to the *political* right.**

However, Reddit post popularity operates on "rich-get-richer" mechanisms [18], where upvotes result in more exposure, more upvotes, and so on. This could mean that CMV participants are less able to influence their communities' norms. Thus, a potential problem that de-polarization scholars can examine is not the ostracization of these users per se, but rather, finding ways of making their (already accepted) voices more influential within their communities.

**Costs of being moderate.** With regards to **RQ2**, CMV participants' language is more personable and advanced and less hostile and confident, with less abstract topical foci. Given that extremity is rewarded in some communities [13], this might mean that more

moderate and friendlier language puts users at a disadvantage in receiving social rewards. Thus, a potential risk is that users may be motivated to be more extreme and appear more confident in order to receive more approval from their peers, which can harm the quality of discussions taking place within the community.

Overall, this presents a challenge in that it may be moderation itself that attracts fewer social rewards, but once again, it is not *penalized* per se. In light of this, it may be more fruitful to examine pathways of making such language normative over the longer term, rather than immediately attempting to introduce it within communities.

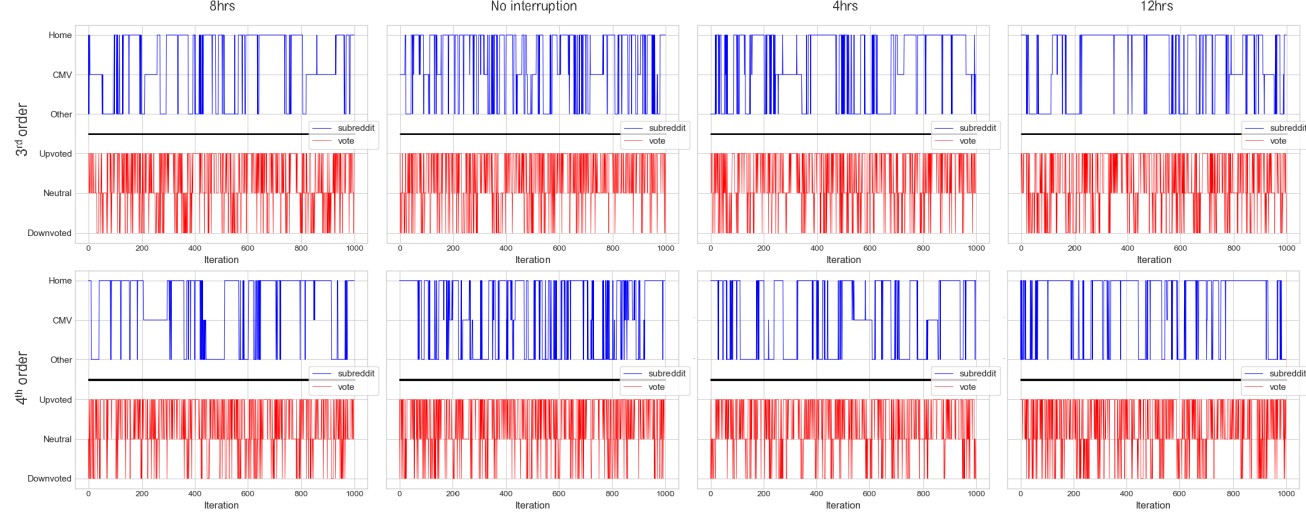

**Figure 3: 1000-comment Markov chain simulations with all configurations. Community fluctuations are higher with no interruption as each new session's starting community is picked randomly. Unsurprisingly, users spend most of their time at home.**

**Social approval in discussion communities.** For **RQ3**, users mostly tend to stay in their current communities in the short term, regardless of their comments' scores. However, it is unclear whether these scores have an effect over the longer term.

These findings show that users are keen to "stand their ground," which is generally optimistic; CMV participants are, at least in the short term, resilient to disapproval of their views, which is an important aspect of online deliberation. Discussions between such users are unlikely to be cut short simply due to perceived disapproval, which may allow the conversing parties to adopt new views. This is antithetical to findings from prior work [7, 23], although this may be expected as we are specifically focusing on a community where disagreement is welcomed.

## 7.2 Limitations and Future Work

Our work is not free from some limitations which we discuss here, along with how these could be rectified in future work.

**Causality and confounders.** As we observe naturally-occurring phenomena, we cannot ascribe a direction to these effects, e.g., whether it is users' exposure to opposing views that cause their comment scores/sophisticated language or vice-versa, or if it is a third factor driving these patterns. Moreover, inaccessible deleted content in the data could warp our findings. Strictly controlled experiments, established precedence of events, or user-reported data alongside digital trace data may be required in future work.

**User open-mindedness.** Though CMV participants are likely to be open-minded, this does not *necessarily* mean that non-participants are the opposite. Future work could employ other methods of classification; for example, overall bias in the news links that users provide [12] (keeping in mind that some links may be shared disingenuously), or detecting and studying more discussion subreddits.

**What kind of engagement?** For our user trail analyses, we were mostly concerned with *whether* the users continued to engage in CMV, but not the *quality* of such discussions. Future work could explore this further, by examining whether users may become more subjective, hostile, or negative following a downvote even within these otherwise open-minded communities.

## 8 ETHICAL STATEMENT

This work has undergone ethical review and received approval by the authors' institution. We only use data that is in the public domain, and do not attempt to de-anonymize or track users. The institution name and project ID are omitted due to anonymity purposes but will appear in the camera-ready version of the manuscript.

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

## A BOT REMOVAL

To detect bots, we obtain all of the comments posted by each of the users in our matched pairs with at least 10 comments during the observation period. We then compute the pairwise cosine similarity between the bag-of-word vectors of each comment for each user, such that every user has their own similarity matrix. Other than tokenization, we do *not* pre-process the text to retain the exact tokens that bots repeat in their comments. Thus, our approach is only suitable for overt bots. We plot the Cumulative Distribution Function (CDF) of user average comment similarity in Figure 4. We focus on the small subset of users with similarity values ≥ 0.4 as nearly all users are under this threshold, leaving 410 users.

To determine the similarity cutoff for bots, we rely on a heuristic name-based rule. We label users as bots if their usernames contain any bot-like word (bot, auto, moderator), or any popular platform name (Reddit, YouTube, Facebook, Twitter, Imgur), as many bots provide services related to these platforms and thus mention them in their names. We perform a logistic regression with class weighting on user text similarity, acknowledging that some bots may not have the heuristic words in their usernames and vice-versa. We take the similarity point at which predictions switch from non-bot to bot as our cutoff (0.59). Model predictions against heuristic annotations result in an F1 score of 0.85.

We treat any users who exceed the cutoff as bots, obtaining 111 accounts. Further, we manually check whether the 28 users with bot-like naming conventions below the cutoff appear in bot detection subreddits (e.g., r/BotDefense, r/BotTerminator, etc.), state they are bots in comments/profile descriptions, or show suspicious behavior. Of these accounts, we find 26 more bots, bringing the total to 137.

## B READABILITY FORMULAS USED

(1) **Flesch-Kincaid Grade Level:** Relies on total syllables, words, and sentences in text.
(2) **Flesch Reading Ease:** Relies on total syllables, words, and sentences in text. Uses different weighting than Flesch-Kincaid Grade Level.
(3) **SMOG Index:** Relies on number of polysyllables (i.e., words with ≥ 3 syllables) and number of sentences.
(4) **Coleman-Liau Index:** Relies on average number of words and average number of sentences per 100 words.
(5) **Automated Readability Index:** Relies on number of characters, words, and sentences in text.
(6) **Dale-Chall Formula:** Relies on the ratio of "difficult" words to total words, and the ratio of total words to total sentences. Difficult words are those that appear in a curated list.
(7) **Linsear Write Metric:** Relies on a point system based on number of syllables in each word and total number of sentences.
(8) **Gunning Fog Index:** Relies on number of words, polysyllables, and sentences.

## C FOLLOW-UP DOWNVOTES ANALYSES

Here, we show complementary analyses to Section 6.

### C.1 Early Sanctions in CMV

We consider the role of each user's *first* comment in CMV as this may have a disproportionate impact [23], especially considering the challenges faced by newcomers when entering new communities [39] and the negative impact on content quality when excluding them [16]. We compare users who only have a single comment in the community during our observation period to those who have more than one, since the former may have left due to early perceived disapproval of their views.

We count downvoted and non-downvoted comments among these users, and compare them to 1) the total pool of comments in CMV throughout 2018 and 2) the pool of first comments by users who went on to post more. Chi-squared tests reveal the opposite pattern: downvoted comments among one-time commenters are fewer compared to both the total pool ($\chi^2(1) = 131.37$, $p < 0.001$) and the first-comment pool of other users ($\chi^2(1) = 295.83$, $p < 0.001$).

This somewhat agrees with our earlier short-term analysis in that users may "stand their ground" in CMV. This pattern could be due to multiple reasons; for example, users who have their first comments downvoted may be further motivated to make others see their point of view, or they may comment in highly contentious discussions, which may overall attract more engagement [9].

### C.2 Long-Term CMV Residents and Departees

To examine whether downvotes which are consolidated and built up over the longer term play a role in whether a user leaves a discussion community, we separate users into those who stayed active in CMV throughout the year, and those who left at some point. We sample users who posted at least 10 comments in CMV between January 1st - June 30th, 2018 (i.e., the sampling period). From this, we subset those who also posted at least one comment in CMV between October 1st - December 31st, 2018 (i.e., the long-term residents). The remaining subset reflects the departees. The "dead period"

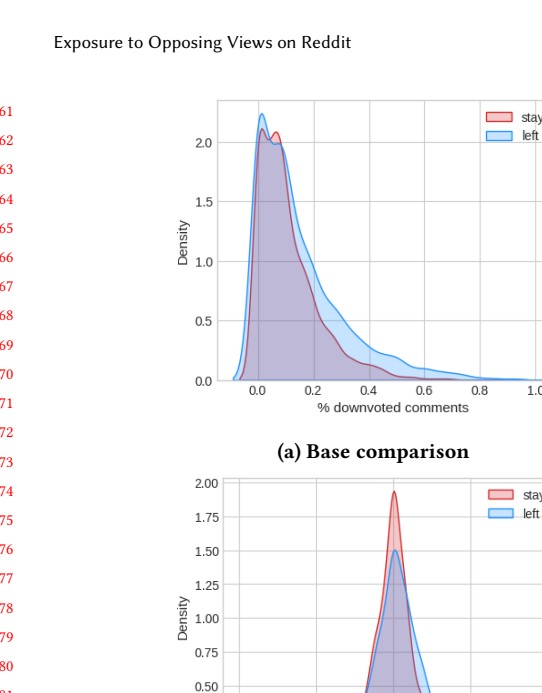

**(a) Base comparison**

**(b) Comparison to home**

**Figure 5: Comparison of percentage downvoted between long-term residents and departees.**

(July 1st - September 30th) is to allow for consolidation of comment votes during the sampling period.

Then, we compare the proportion of *downvoted* comments during the sampling period between these two sets using independent t-tests. Indeed, we find that downvotes are more prominent among departees ($N$ = 4678, $M$ = 15%) than long-term residents ($N$ = 3549, $M$ = 11%), $t$ = 12.54, $p < .001$, Cohen's $d$ = 0.279 (small effect size), indicating that downvoting may be a factor in their eventual departure.

This pattern also holds when examining relative downvotes compared to what users receive at their homes. We compare the *difference* in percentages of downvoted comments between each user's CMV and home comments, omitting anyone who has CMV as their sole home. This confirms that departees are more downvoted in this community relative to their homes ($N$ = 3033, $M_{difference}$ = 4.63%) than long-term residents ($N$ = 2931, $M_{difference}$ = 0.84%), $t$ = 8.95, $p < .001$, Cohen's $d$ = 0.232 (small effect size).

However, looking at the distributions of each group of users with respect to downvote percentages in Figure 5, we see that these differences are mostly driven by the right ends of the curves, with substantial distribution overlap. In fact, most users in both groups have no downvoted comments. Thus, our results here are inconclusive as there may be several factors behind user departures from the subreddit.

Received 12 October 2023; revised XXX; accepted XXX

