# OpenReview forum: "Deliberate Exposure to Opposing Views and its Association with Behavior and Rewards on Political Communities"
_ACM.org/TheWebConf/2024/Conference — TheWebConf24_

### Official Review · Reviewer_uif2 · 2023-11-19

**Novelty:** 4
**Technical Quality:** 5

**Review:**

[Summary]

This paper analyzes users who are active in one of four political subreddits across the political spectrum (r/Conservative, r/The_Donald, r/ChapoTrapHouse, r/SandersForPresident) and have also posted or commented in r/changemyview (CMV). It considers the language in their posts, the social rewards, and the social punishments they receive in their home communities. It finds that (a) their posts use more personable and less hostile language, (b) they receive fewer social rewards (upvotes), and (c) they do not receive as much social punishment (downvotes) from their peers as to drive them out of their home community.

[Strengths]

- The paper is well-written and easy to follow.
- The research questions are relevant, well-posed, and interesting.
- I appreciated the careful selection and application of statistical methods in various parts of the paper.

[Weaknesses]

- The analysis in the paper treats the same way users who have posted and users who have commented in CMV. Those two groups feel substantively very different to me: one group is openly asking to be convinced to change their views, while the other tries to convince others to change their views. One may argue that mere engagement in CMV is enough, but it’s also not hard to imagine that users who are entrenched in their views would come to CMV to convince others to adopt their views. I wonder how the findings presented in the paper would change if these two groups were analyzed individually.

- The paper is a bit thin on interpretation of the results. The results are simply reported and not put into context. For example, Sections 4.3 and 5.1 both report that the effect sizes are small, but what do these effects mean in substantive terms? I mentioned these two examples, but I had the same impression in other parts of the paper.

- While I did appreciate the careful statistical treatment (as I mentioned in the strengths), I did feel that it got a bit on the way of communicating the substantive importance of the results. I know that this is a hard balance to strike, i.e., explaining that you were careful vs. explaining the results and their implications.

- User matching (4.1): why were these particular characteristics chosen to match participants? The current set focuses on the volume of activity, the proportion of activity in the home subreddits, and engagement in other subreddits, but does not, for example, include any features related to the kind of content the users post/engage with. It would be useful to hear the authors’ reasoning behind choosing the matching features and the limitations of the decision. I’m aware that any matching procedure is “imperfect”, but given that the matching influences all subsequent analyses, I think it’s important to articulate that decision better.

- User matching (4.1): while it’s useful to see the Standardized Mean Difference in Table 2, it would be even more informative to know whether these differences are statistically significant. One way to test this is to fit a model that predicts whether the users in “case” or “control” as a function of the matching features and to check whether any of the matching features are statistically significant in that model.

- The analysis in Section 6 of the effects of downvotes on users’ future engagement in their home communities is a bit unclear and overly complicated. There are simpler, more interpretable analyses to get at this question than fitting a 4th order Markov chain. I’m not against a more complex approach when it adds to the analysis, but it didn’t seem that way here. For instance, are these just results for cases? Why not compare the responses to the home community feedback of cases vs. controls in terms of engaging in the home community vs. other communities?

I have read the authors' responses.

**Questions:**

See above.

**Reviewer Confidence:**

4: The reviewer is certain that the evaluation is correct and very familiar with the relevant literature

**Scope:**

3: The work is somewhat relevant to the Web and to the track, and is of narrow interest to a sub-community

---

### Official Review · Reviewer_DR9h · 2023-11-22

**Novelty:** 4
**Technical Quality:** 4

**Review:**

Positive aspects of the study:
- Theme is relevant and timing
- Use of real large-scale data
- Promising ideal

Negative aspects:
- Lack of formality. Some key parts of the text lack formality, for instance, regarding the procedures of obtaining the studied groups and the considered matching – section 4.1. This makes important details not clear.

- Lack of clarity: Authors use a lot of acronyms, making several parts of the text difficult to follow. The presentation of the results could also be better organized. In some parts, results are presented without discussing properly supporting evidence to the readers. Some other examples of lack of clarity: “Next, we compare the average scores between..” what scores? Some results are not presented in the best way—for instance, the perspective features. The authors do not show the distribution or mean/median values; what do they represent?

- Lack of enough evidence. Some key affirmations are not properly discussed, and the reader is not provided with enough evidence. For instance, “Altogether, CMV participants receive fewer social rewards than non-participants.”

- Methodological problems. E.g., see points listed below.

**Questions:**

Are the mean source bias per group per subreddit statistically significantly different? i.e., can we say, for instance, that SFP: -0.73 (Mcase) -0.78 (Mcontrol) and -0.80 ( Mrandom) are statistically different?

How is the new model for NER trained? What does the F1 mentioned refer to? Important details are missing.

We label users as bots if their usernames contain any bot-like word (bot, auto, moderator), or any popular platform name (Reddit, YouTube, Facebook, Twitter, Imgur), as many bots provide services related to these platforms and thus mention them in their names.

Several details are not explained; for instance, what are %diff in table 5?

Methodological problems. For example: “We label users as bots if their usernames contain any bot-like word (bot, auto, moderator), or any popular platform name (Reddit, YouTube, Facebook, Twitter, Imgur), as many bots provide services related to these platforms and thus mention them in their names.” Did you evaluate? Did you contrast the with state-of-the-art methods?

**Ethics Review Description:**

-

**Reviewer Confidence:**

4: The reviewer is certain that the evaluation is correct and very familiar with the relevant literature

**Scope:**

4: The work is relevant to the Web and to the track, and is of broad interest to the community

---

### Official Review · Reviewer_Nu6x · 2023-11-22

**Novelty:** 4
**Technical Quality:** 4

**Review:**

Introduction
●	The introduction motivates the problem space around polarization and importance of understanding users who engage across ideological divides.
●	It would be beneficial to briefly mention the expected effects of social rewards/sanctions on user engagement, considering this is the primary research question.

Related Work
●	A notable limitation is the absence of directly relevant studies focusing on users who actively break out of polarized communities. Who are these people? Why would they do that? More closely related work on minority opinion holders and effects on their engagement could strengthen this section. Brokerage seems to be a hidden important term here.

Dataset
●	Appropriate datasets collected to address the questions. Breadth across the political spectrum is a plus.
●	The definition of a 'political subreddit' remains unclear. What constitutes “political comments”? How do we know if a subreddit contains over 50% political comments? Do these subreddits have more user overlap with CMV than others?
●	More justification is needed (in the Introduction as well) on looking at only CMV as an “anti-echo chamber” and not other debate spaces. What makes CMV truly special, especially compared to ones like r/PoliticalDiscussion?
●	Although there are the numbers of overlapping users (“cases”) in Table 2, it would be better to have a separate table that shows more about the descriptive profiles of “cases”, as these users are the main focus of this paper.

Results
●	While the detailed reports of results, supplemented with numerous tables and figures, are adequate, referencing specific parts proves challenging. Better visual representation of the results are welcome.
●	There's a lack of focus between RQ1-2 and RQ3. Section 4-5 analyzed how CMV users who also belong to either TD, CTH, CON, or SFP behave and are rewarded. The logical next step is to show if they stay in their home community (TD, CTH, CON, or SFP) and be more or less influential in their home. I’m more curious about if CMV people stay in those 4 political subreddits despite the social sanctions they face in those subreddits. If they still stay, there might be some spillover effect of depolarizing the 4 subreddits. If they don’t, it would be yet another sign of polarization. As the paper focuses on the role of the users that are willing “to partake in heterogeneous discussions” and to bring “new ideas into these groups”, I recommend adjusting RQ3 and the analysis accordingly.
●	As mentioned in the paper, almost all differences between control and case groups have small effect sizes. Talking through whether any of these small effect sizes still have impact would be helpful.
●	“Though we expected CMV participants to also use more analytical language” <- Why would this be expected? Also, if the results are truly counterintuitive, can the claim “because formal language is more normative within the political spaces we study” be backed better?
●	“CMV participants in this [CTH] subreddit use more swear words” <- There’s no further discussion on this major counterexample. What factors would bring this reversed relationship, compared to others? Is it because of some of the properties of Perspective API, and/or CTH?
●	There’s not enough justification on the use of Perspective API and LIWC. Do these models fit well to the Reddit corpus?
●	Table 6 reports the proportion of retained text. Would retaining only <5% of texts for readability analysis yield reliable results? Is this truly inevitable?

Update: I ack reading the authors' responses and considered making score changes as deemed appropriate.

**Questions:**

Concentrate on aligning the focus of the Research Questions (RQs) more coherently.
Incorporate additional related works for a more comprehensive literature background.
Refine the methodology explanation to enhance clarity and justification.
Improve data presentation for better accessibility and impact.
Deepen the discussion to more thoroughly explore the implications and context of your findings.
What implications does looking at these users have on other platforms or on the society in general?

**Reviewer Confidence:**

4: The reviewer is certain that the evaluation is correct and very familiar with the relevant literature

**Scope:**

3: The work is somewhat relevant to the Web and to the track, and is of narrow interest to a sub-community

---

### Official Review · Reviewer_BJBy · 2023-11-23

**Novelty:** 6
**Technical Quality:** 6

**Review:**

The paper addresses some unexplored questions regarding users who participate in deliberative discussions that welcome diverse opinions. The operationalization of this specific group – users from political communities who also comment in the CMV community – is an interesting approach. The employment of various methods, including matching, toxicity detection, topic extraction, and higher-order Markov chains, adds depth to your analysis. This paper could benefit from a closer alignment of its empirical analysis with the identified literature gaps, specifically by examining rewards variance and downvotes. More clarification in Named Entity Recognition (NER) usage, and a deeper exploration into the unexpected use of less analytic language by CMV participants are also needed.

**Questions:**

Your paper delves into an underexplored topic with a solid design and sophisticated methods. However, I have a few suggestions to enhance its quality and clarity:

Connection Between Literature and Empirical Analysis: While you identify a literature gap concerning whether individuals from this group are penalized by their communities, the empirical analysis seems to focus more on the comparison of upvotes than downvotes. Testing whether CMV participants receive fewer rewards is a logical step, but examining the variance in rewards across the four communities could yield more insights.

User Engagement Analysis: It would be beneficial to provide statistics on how many users in your case group comment only once and the distribution of their engagement. This data would help understand the extent of genuine participation in diverse communication among these users.

Rationale for NER Use: The purpose of employing Named Entity Recognition (NER) in your study is not entirely clear. Are there specific reasons to anticipate differences in the entities mentioned by the case and control groups? Clarifying this would help justify the inclusion of NER in your methodology. The space saved here could be used to expand on the discussions in Section 6.

Psychological Trait Analysis: The finding that CMV participants use less analytic language than expected is intriguing. Further exploration and potential explanations for this observation would be valuable. Presenting examples of analytic language used by the control group and personal language by the case group could offer more clarity.

**Reviewer Confidence:**

4: The reviewer is certain that the evaluation is correct and very familiar with the relevant literature

**Scope:**

4: The work is relevant to the Web and to the track, and is of broad interest to the community

---

### Official Review · Reviewer_Y75p · 2023-11-24

**Novelty:** 5
**Technical Quality:** 5

**Review:**

This paper investigates the dynamics of online political discourse, particularly focused on users who engage with diverse political views while rooted in ideologically homogeneous online communities (Reddit). The study's examination of four distinct communities (r/Conservative, r/The_Donald, r/ChapoTrapHouse, and r/SandersForPresident) provides an overview of user behavior across a political spectrum.

Pros
- This is an important and timely topic, and the paper highlights and engages the challenges faced by many efforts for political "de-polarization"
- The findings ("open-minded" users tend to employ less hostile, more advanced, and personable language yet receive fewer social rewards from their peers) are illuminating
- The sampling/data collection strategies are sound
- Measurements (e.g., ideology (Section 4.2.)) and strategies taken to deal with confounding (like matching for the case users and the control users) are effective

Cons
- While the authors examine topics (p. 3, line 311) as one of the many linguistic differences, I wonder if this might be a factor that drives differences in the other linguistic dimensions (e.g., hostility attributes).
- I am unconvinced by some of the decisions on data transformation, including trimming the top 5% values as outliers (to my knowledge, there would some resampling strategies to consider as an alternative)

**Questions:**

I wonder how confident we are in the degree to which the arguments in CMV are made "genuinely", posting users must "truly" be willing to change their view, and they want to expose them selves to diverse views in a "civil" and "genuine" manner. Aren't these assumptions, rather than proven facts, that are being investigated by the authors or further need to be vetted?

**Ethics Review Description:**

-

**Reviewer Confidence:**

3: The reviewer is confident but not certain that the evaluation is correct

**Scope:**

4: The work is relevant to the Web and to the track, and is of broad interest to the community

---

### Decision · Program_Chairs · 2024-01-22

**Decision:**

Accept

**Comment:**

Reviewers commended the authors on the timeliness timeliness and significance of the paper's topic. They pointed out the need for better alignment between research questions and analysis, and made numerous suggestions to improve analysis, including discussion of small effect sizes, clarification of linguistic analysis of communities, exploration of counterexamples, justification for the use of certain tools, and addressing concerns about readability analysis. Authors provided detailed response and improved technical aspects of the paper to reviewer satisfaction.